# Validation of a Novel Zebrafish Model of Dengue Virus (DENV-3) Pathology Using the Pentaherbal Medicine Denguenil Vati

**DOI:** 10.3390/biom10070971

**Published:** 2020-06-28

**Authors:** Acharya Balkrishna, Siva Kumar Solleti, Sudeep Verma, Anurag Varshney

**Affiliations:** 1Drug Discovery and Development Division, Patanjali Research Institute, NH-58, Haridwar 249 405, Uttarakhand, India; pyp@divyayoga.com (A.B.); siva.kumar@prft.co.in (S.K.S.); sudeep.verma@prft.co.in (S.V.); 2Department of Allied and Applied Sciences, University of Patanjali, Patanjali Yog Peeth, Roorkee-Haridwar Road, Haridwar 249 405, Uttarakhand, India

**Keywords:** dengue virus, DENV, zebrafish model, gene expression, Denguenil Vati

## Abstract

Dengue is a devastating viral fever of humans, caused by dengue virus. Using a novel zebrafish model of dengue pathology, we validated the potential anti-dengue therapeutic properties of pentaherbal medicine, Denguenil Vati. At two different time points (at 7 and 14 days post infection with dengue virus), we tested three translational doses (5.8 μg/kg, 28 μg/kg, and 140 μg/kg). Dose- and time-dependent inhibition of the viral copy numbers was identified upon Denguenil Vati treatment. Hepatocyte necrosis, liver inflammation, and red blood cell (RBC) infiltration into the liver were significantly inhibited upon Denguenil treatment. Treatment with Denguenil Vati significantly recovered the virus-induced decreases in total platelet numbers and total RBC count, and concomitantly increasing hematocrit percentage, in a dose- and time-dependent manner. Conversely, virus-induced white blood cell (WBC) counts were significantly normalized. Virus-induced hemorrhage was completely abrogated by Denguenil after 14 days, at all the doses tested. Gene expression analysis identified a significant decrease in disease-induced endothelial apoptotic marker Angiopoetin2 (*Ang-2*) and pro-inflammatory chemokine marker *CCL3* upon Denguenil treatment. Presence of gallic acid, ellagic acid, palmetin, and berberine molecules in the Denguenil formulation was detected by HPLC. Taken together, our results exhibit the potential therapeutic properties of Denguenil Vati in ameliorating pathological features of dengue.

## 1. Introduction

Dengue is an important mosquito-transmitted viral hemorrhagic fever of humans caused by the infection of dengue virus (DENV) primarily in tropical and subtropical countries [1]. As per the Center for Disease Control, USA, dengue is endemic in more than 100 countries, with an estimated 400 million infections each year [2], of which approximately 100 million people get sick from infection [3]. It has been predicted that more than 40% of the world’s population are at risk of dengue infection [4]. With the rapid spread of DENV, which is now prevalent in Asia, Africa, and the Americas, dengue has become a major public health threat worldwide [5].

DENV is an enveloped positive-sense ssRNA virus of the Flaviviridae family [5]. There are four serotypes of the virus exist (DENV-1, 2, 3, and 4) [6] and all elicit a similar range of disease manifestations during infection [7]. The first infection with dengue virus typically results in either no symptoms or a mild illness that can be mistaken for the flu or another viral infection. Dengue viral infection causes a spectrum of illnesses ranging from asymptomatic to causing mild dengue Fever (DF), joint pains, rashes, and other mild symptoms [8]. The feature of DF includes vascular leakage, thrombocytopenia, and coagulopathy [9].

Upon inoculation, DENV is thought to infect Langerhans cells, and then migrate through the lymphatic system and the infection spreads to various cells of hematopoietic lineages, including macrophages, monocytes, before spreading systemically [10]. Liver is an important target for DENV infection [11]. DENV induces both innate and acquired immune responses upon infection. Innate immune response, includes complement and type I Interferon (IFN) [12] and acquired immune response involves serotype-specific CD8+ and CD4+ T cells, lysis of DENV-infected cells and production of various cytokines, such as IFN-γ, TNF-α, CCL3, and lymphotoxin [6,13,14].

Development of an appropriate animal model for DENV infection has been hindered by lack of replication of DENV clinical isolates in wild-type (WT) mice; and the lack of clinical disease in nonhuman primates. Nonhuman primates can sustain DENV replication even after inoculation with the presumed mosquito inoculation in humans [15] and occasionally display only low platelet counts and no other overt clinical signs of DENV infection [16], such as fever, anorexia, or lethargy [17]. Further, mouse models of dengue did not show human clinical signs of dengue viral infection, but developed neurotropic disease, which is not generally observed in humans [18]. While few strains of mice support minimal replication of DENV, other strains show signs of paralysis and DENV infection induces limited DENV pathogenesis, such as liver damage, increased white blood cell counts, and thrombocytopenia [19]. Further, lack of a suitable small-animal model of dengue infection has greatly hindered the study of dengue pathogenesis and the development of novel therapeutics [20,21].

Zebrafish (*Danio rerio*) is an excellent model organism to investigate complex biological processes [22]. Besides traditional host models such as mice, zebrafish offers a well-developed immune system with both innate and adaptive immune responses. Further, its remarkable similarities to the human system and signaling pathways involved in the immune response has made zebrafish a greatly used model to research host–pathogen interactions, including viral infections and antiviral drug screening, hence providing original therapeutic avenues [23]. Further, use of zebrafish for disease modelling of rare genetic blood diseases and for drug discovery in Parkinson’s disease and other movement disorders is apparent [24,25]. In the context of viral infections, zebrafish models have unexpected idiosyncrasies among organs that may apply to the human situation [22].

Imbalance of host response to the infection results in severe dengue. Currently, there is no licensed therapeutic drug available to treat dengue [26]. Therapeutic modulation to inhibit or mitigate the progression of the disease has been investigated for many decades. However, these intensive efforts to develop antiviral drugs as well as clinical trials using repurposing drugs or with natural products to prevent the plasma leakage have not been encouraging so far. While the development of vaccines has been held back for a while due to the lack of a good animal model and the complexity of disease [27], the FDA has recently approved Dengvaxia^®^ [3], the only available vaccine against dengue.

Despite being an old disease, therapeutic options available for prevention and control of DENV infection are severely limited. Hence, developments of cost effective, efficacious prophylactic/therapeutic drugs against DENV are needed. Many plant-derived herbal medicines and purified natural compounds have been shown to exert antiviral effects in animal studies and human clinical trials [28]. Natural products are a potent source of combinatorial chemicals as they contain structurally diverse bioactive chemicals, thus making them a valuable resource for novel drug discovery. Natural products and their derivatives accounted for 34 % of new medicines approved by the US Food and Drug Administration (FDA) between 1981 and 2010 [29]. Further, natural products are more likely to reach the site of action and have better coverage of biologically relevant chemical space [29,30]. Uses of several natural products against dengue viral infection have been reported [31,32,33].

Denguenil Vati is a pentaherbal Indian traditional formulation that has been prescribed by ayurvedic practitioners for the treatment of dengue infection in India. However, its anti-dengue viral properties have not yet been explored scientifically. Denguenil consists of extracts from *Tinospora cordifolia* (Willd.) Miers, *Aloe vera* (L.) Burm.f, *Carica papaya* L., *Punica granatum* L., and *Ocimum sanctum* L. The aim of the present research work is to validate potential antidengue therapeutic properties of Denguenil in vivo using the zebrafish model of disease. In addition, HPLC analysis was undertaken to quantify chemical fingerprints present in the Denguenil and their correlation with observed biological responses.

## 2. Materials and Methods

### 2.1. Ethics Statement

All the animal experiments were performed according to ethical guidelines as per the Committee for the Purpose of Control and Supervision of Experiments (CPCSEA), Government of India; and the established zebrafish protocols were approved (IAEC study number-218/Go062019/IAEC) by Institutional Animal Care and Ethics Committee. Informed written consent was obtained from all patients. The collection and storage of patient serum samples were also approved by the institute’s ethical committee (Approval number-218/IEC/062019/Pent/TN). All methods were carried out in accordance with relevant guidelines and regulations of the relevant ethical committee.

### 2.2. Zebrafish Care and Maintenance

Zebrafish were maintained in the dedicated zebrafish research facility according to IAEC standards. IAEC approved guidelines for zebrafish care followed the standard procedures of a 14 h light, 10 h dark cycle at 28 °C. Zebrafish of similar bodyweight were selected for the experimental study and were housed in polycarbonate tank at a stocking density of 2 liters of water per fish.

### 2.3. Propagation of Dengue Virus (Serotype, DENV-3) in Zebrafish

Human serum samples were collected with informed patient consent at the source clinic and transferred to the laboratory for the study. The samples were obtained from a total of three male dengue-positive subjects, within age groups of 21 to 45 who had fever accompanied with joint and muscle pain. All the patients confirmed to not having other comorbid conditions. Fresh serum samples from these dengue-positive patients (Serotype, DENV-3) were maintained at 4 °C under controlled conditions and were used within 96 h of collection. Briefly, samples were brought to room temperature in a water-bath just before use. Zebrafish were anesthetized by placing them in ice water (18 °C). The fish were held intact with a wet sponge, and an aliquot of 3 µL dengue-infected human serum was injected intramuscularly into the proximo-distal region of adult zebrafish (n = 10) as primary carrier and to propagate the virus. Fish receiving same volume of 0.1 % saline served as control. After injection, fish were transferred to water tank for recovery. After 14 days of viral induction, 1 µL each of serum samples were harvested from five primary carriers and diluted to 100 µL with sterile PBS for the study. From this, 3 µL of diluted serum from primary carriers were injected into the study fish.

### 2.4. Blood Collection Harvesting Serum from Carrier Zebrafish

Fish were euthanized in ice water and the fish surface was wiped dry. A sharp slit was made between the anal fin and the caudal fin region, thus isolating the caudal fin, and the fish was held with the wound facing down. Whole blood was collected at the dorsal aorta using a P20 micropipette fitted with an elongated tip and aspired into prechilled microcentrifuge tubes. Both pipette tips and tubes were precoated with EDTA by submerging in 18 mg/mL EDTA solution for 24 h and then dried prior to use. The samples were centrifuged with 2% EDTA at high speed of 7000 rpm for 10 min. Following centrifugation, the serum was collected very carefully from the top layer, without disturbing the layers.

### 2.5. Dengue Viral Infection Study Design

After viral infection, the study groups (n = 24) were cotreated with Denguenil from day 7 for effective dose (ED) screening. Two time points were chosen for the effective dose screening after 7 and 14 days of Denguenil dosing (i.e., on the 8th day and 15th day) of viral-serum-infected zebrafish. Virus-infected zebrafish receiving serum plus PBS served as the disease control; fish receiving only PBS without any virus served as the normal control.

### 2.6. Preparation of Zebrafish Test Feed and Dosing

The study compound, Denguenil, was mixed and ground with a known volume of fish feed, then extruded to pellets of standard size weighing 4 mg per pellet for dosing. A 24 h feeding cycle was followed throughout the study period. Test article information for each study group was blind coded for investigators and fish-handlers. In the present study, Denguenil was crushed and diluted with PBS. For preparing fish feed, the required volume of Denguenil was mixed with feed and extruded. For control group, fish feed was mixed with an equal volume of PBS without drug and extruded. Hence, fish feed containing Denguenil was used to treat disease model, whereas fish receiving normal feed without Denguenil served as the control.

### 2.7. Harvesting Liver Tissue and Caudal Fins for Histopathology

On the last day of the experiment, zebrafish were euthanized with cold water and an incision was made in the viscera to harvest the livers. Livers were divided into two parts, for RNA isolation and for cell smear preparation. For RNA isolation, liver tissues were frozen in liquid nitrogen immediately after sectioning and stored at −80 °C till further use. The caudal fin isolated was mounted on a slide for further observation.

### 2.8. RNA Isolation and Gene Expression Analysis

The total RNA was extracted from fresh tissue using RNAqueous^®^-Micro Kit (Thermo Fisher Scientific, Waltham, MA, USA) as per the manufacturer’s instruction and rendered DNA free, quantified, and subsequently storing at −80 °C. cDNA synthesis was performed using SuperScript II reverse transcriptase (Thermo Fisher Scientific, Waltham, MA, USA) employing Oligo(dT) primers and the target cDNA templates were amplified by PCR using gene specific primers with 2X PCR Master mix (Thermo Fisher Scientific, Waltham, MA, USA). After 40–50 cycles, the amplified products were quantified by mean absorbance using a standard 10 nanomoles reference. A total of 200 µL of QuantiFluor^®^ ONE dsDNA System (Promega Corporation, Madison, WI, USA) was added to 10 µL of amplified cDNA, incubated at room temperature for 5 min and readings were made at (504 nm Ex/531 nm Em) using a Robonik microplate reader. The following primers for zebrafish sequences were used for the study.
DENV-3:Forward 5′-CGGGAAAACCGTCTATCAATATGC-3′Reverse 5′-TGAGAATCTCTTCGCCAACTGTG-3′CCL3:Forward 5′-CCGCGGATCCGACGATTTA-3′Reverse 5′-AATGACTCCAGGCAGAGTGC-3′ANG2:Forward 5′-TATTTGTGAGGTTTTCCGTTCCCATCGGGCT-3′Reverse 5′-AGAGGACTATGAGAAGTCGGCTCCTCGGATCAT-3′.

### 2.9. Complete Blood Count

For complete blood analysis, whole blood was collected from each fish separately using heparinized microhematocrit tubes, and samples were centrifuged at 14,000 rpm for 13 min, resuspended in 1 mL phosphate buffered saline (PBS), and counted using a hemocytometer, before smears were made and stained with Hematoxylin and Eosin (H&E).

### 2.10. Hematoxylin and Eosin (H&E) Staining, Imaging, and Cell Counting

Fixed glass slides with sample smears were stained with H&E for 2 min each, followed by three PBS washes and allowed to dry at room temperature. Slides were viewed under microscope (Labomed l× 400 microscope) at 45× magnification and bright field images were captured by Future Winjoe image capture system. One hundred cells were counted per field for quantification.

### 2.11. Cell Culture

The cell lines used in this study included human liver cancer cell lines, HepG2 cells, and human epidermoid carcinoma cell lines, A431 cells (purchased from ATCC licensed cell repository, National Centre for Cell Science (NCCS), Pune, India). Both of these cell lines were maintained in DMEM (Dulbecco’s modified Eagle’s medium; Invitrogen, USA) supplemented with 10% FCS and antibiotics. Cells were placed in a humidified incubator at 5% CO_2_ at 37 °C. For in vitro experiments, cells were seeded at 2 × 10^4^ cells/well in 96-well plates and experimental treatments with Denguenil Vati was carried out for 24 h in DMEM + 1% FBS media at 70% confluence.

### 2.12. Cell Viability Assay

HepG2 and A431 cell viabilities were measured using Alamar blue reagent (Hi Media, India). Briefly, both HepG2 and A431 cells (2 × 10^4^ cells/well) were seeded in 96-well plates and treated for 24 h with various concentrations of Denguenil Vati (1–1000 µg/mL). Two hours before termination, 10 µL of Alamar blue (0.15 mg/mL) was added to each well. Cytotoxicity was measured by reading fluorescence at Ex. 540/Em. 590, using an Envision microplate reader (Perkin Elmer, Waltham, MA, USA).

### 2.13. Reactive Oxygen Species (ROS) Measurement

The intracellular ROS levels were quantified by a fluorescent probe, 2,7-dichlorofluorescein-diacetate (DCFH-DA, Sigma, St. Louis, MO, USA). HepG2 and A431 cells were seeded in 96 well plates and treatment with various concentrations of Denguenil Vati (1–1000 µg/mL) for 24 h. Cells were washed with PBS and incubated with DCF-DA (10 µg/mL) in serum free DMEM for 1 h in dark. The ROS production was quantified by measuring DCF fluorescence intensity at Ex. 495/Em. 527 using Envision microplate reader (Perkin Elmer, Waltham, MA, USA). Results were expressed as the percentage of ROS generation, as compared to untreated control cells.

### 2.14. Malondialdehyde (MDA) Estimation

Lipid peroxidation was estimated by measuring the levels of malondialdehyde (MDA) in cell culture spent medium. MDA estimation assay was based on the reaction of MDA with Thio-Barbituric acid (TBA); forming an MDA–TBA adduct [34]. Its absorption at 532 nm was measured using an Envision microplate reader (Perkin Elmer, Waltham, MA, USA) and normalized with untreated control cells.

### 2.15. Preparation of Denguenil Sample for HPLC

Denguenil was sourced from Divya Pharmacy, Haridwar, India (Batch number A-DNV014). Tablets were crushed into fine powder; 10 mL methanol was added to 0.5 gm of powdered sample; and sonicated (Thermo Fisher Scientific, Waltham, MA, USA) for 30 min. The samples were centrifuged at 8000 rpm for 5 min; filtered using 0.45 µm nylon filter (Sol A); and used for the analysis of chemical signatures present. The samples were analyzed using gallic acid, and ellagic acid (Sigma Aldrich); palmetin and berberine (Natural Remedies, Bengaluru, India) as reference standards with purity ≥ 98.0%.

The quantification of signature compounds was performed using High Performance Liquid Chromatography (Waters Corporation, Milford, MA, USA) equipped with binary pump (1525), PDAD (2998) and autosampler (2707) using a Shodex C18-4E (4.6 mm ID × 250 mm L) column at a flow rate of 1.0 mL/min The filtered solution (Sol A) was used for the analysis of palmetin and berberine. Next, 2.0 mL of Sol A was diluted to 5 mL with same solvent and used for the analysis of ellagic acid and gallic acid. The chromatographs were recorded at 270 nm (gallic acid and ellagic acid) and 346 nm (palmetin and berberine) wavelengths.

### 2.16. Statistical Analysis

All in vivo experiments were conducted with multiple replicates (n = 24), and in vitro experiments were conducted at least thrice in triplicates and data are expressed as means ± SEM. Comparisons between experimental groups were made by one-way ANOVA, followed by Dunnett’s multiple comparison post hoc test. Differences in mean values were considered significant at *p* < 0.05. Significance of data were analyzed using GraphPad Prism 7.03.3.

## 3. Results

### 3.1. Study Design for Translational Dosing

Dengue is a mosquito-borne viral fever of humans with limited therapeutic options [26]. In the present investigation, we screened for the translational effective dose (ED) of ayurvedic herbal formulation, Denguenil. Considering the body weights and body surface area of adult zebrafish and human, the translational dose of Denguenil was determined to be 1000 times less than the relative human dose, as described [35,36]. Human serum infected with dengue virus (DENV-3) was propagated in primary carrier fish. After 14 days, the serum was harvested from carrier fish, diluted, and injected into study fish. A week after injection, the study fish were cotreated with Denguenil-impregnated feed until the termination of the study (Figure 1A). At first, 50% tissue culture infective dose (TCID50) for DENV-3 was titrated using an end point neutralization assay employing VeroE6 cell lines and calculated using the Reed–Muench method to be 420 TCID50/mL. For induction of the disease model, 1 µL of serum from carrier zebrafish was diluted with PBS to a final concentration of 1 TCID50/mL and a total of 3 TCID50 was used as the induction dose. In the present investigation, fish were fed with the indicated doses individually in separated tanks, and once the dosing was complete, they were put back into the original tank for housing.

In an effort to identify the maximum therapeutic efficacy of Denguenil in dengue model of zebrafish, a wide range of translational effective doses were tested in the present study with 0.2× (ED-1, 5.6 μg/kg), 1× (ED-2, 28 μg/kg), and 5× (ED-3, 140 μg/kg) dose of the Denguenil translated from the human prescribed dose (1000 mg/day, BID); these were used to identify the effective dose with maximum therapeutic efficacy. Further, two time points were chosen for the effective dose screening after 7 and 14 days of Denguenil dosing (i.e., on the 8th day and 15th day) of viral-serum-infected zebrafish (Figure 1A). The course of dengue infection included disease initiation, in which hypoactivity, body discoloration, and RBC infiltration were noticed. The critical progression phase showed vascular hemorrhage, hepatocyte necrosis, elevated expression of DENV-3 viral transcripts, and concomitant increase in Ang-2 and *CCL3* expression.

### 3.2. Denguenil Inhibited DENV-3 Viral Transcript Copy Numbers

We screened for the expression of dengue viral biomarker (DENV-3) in the liver samples. The results indicated an active viral infection in the Disease Control. Dengue viral infection resulted in forty-fold increased DENV-3 transcript copy number (Figure 1B) in both the 8th day group (Normal Control, 0 PDU/mL vs. 40 PDU/mL, Disease Control, *p* ˂ 0.0001) and 15th day group (Normal Control, 0 PDU/mL vs. 40 PDU/mL, Disease Control, *p* ˂ 0.0001), whereas no DENV3 copies were detected in Normal Control. Irrespective of time point, Denguenil administration resulted in a dose-dependent decrease in DENV-3 transcript copy numbers (Figure 1B). While a 58% decrease (40 PDU/mL vs. 17 PDU/mL, *p* ˂ 0.0001) in DENV-3 copy number was detected on the 8th day at 140 μg/kg dose, the copy number further decreased to 72% on the 15th day (40 PDU/mL vs. 11 PDU/mL, *p* ˂ 0.0001) (Figure 1B).

### 3.3. Denguenil Inhibited Dengue Virus Induced Hepatocyte Necrosis

We examined the hepatopathological features using hematoxylin and eosin staining of the liver smears. Control liver smears from both 8th and 15th day group showed normal and intact hepatocytes with well-defined cellular architectures (Figure 2A,F). Dengue viral infection in Disease Control fish resulted in a 39% increase in cell necrosis by the 8th day, with a majority of shrunken cells (Figure 2B). Further, the necrosis was significantly increased to 41% by 15th day with shrunken and darkly stained necrotic hepatic cells (Figure 2G), (Disease Control, 8th day group, 39% vs. 41%, 15th day group, *p* ˂ 0.0001) (Figure 3A). Conversely, Denguenil treatment significantly inhibited the hepatocyte necrosis at both time points in a dose-dependent manner (Figure 2C–E,H–J and Figure 3A). At 5.6 μg/kg Denguenil, a modest decrease in the necrosis was identified on both 8th and 15th days compared to the disease controls (8th day group, Disease Control, 39% vs. 33%, 5.6 μg/kg Denguenil, *p* ˂ 0.0001; 15th day group, Disease Control, 41% vs. 40%, 5.6 μg/kg Denguenil, *p* ˂ 0.05) (Figure 2C,H and Figure 3A). While drug treatment decreased necrosis to 29% on the 8th day (Disease Control, 39% vs. 29%, 28 μg/kg Denguenil, *p* ˂ 0.0001), the necrosis was further decreased to 22% by the 15th day (Disease Control, 41% vs. 22%, 28 μg/kg Denguenil, *p* ˂ 0.0001), suggesting a significant inhibition of hepatic cell necrosis by 45% upon receiving 28 μg/kg Denguenil in the 15th day group (Figure 2D,I and Figure 3A). Denguenil at 140 μg/kg has been found to be the effective dose with maximum protection (86%) from dengue viral damage in the 15th day group. In fish receiving 140 μg/kg Denguenil, while 16% sparsely scattered necrotic cells were identified in the 8th day group (8th day group, Disease Control, 39% vs. 16%, 140 μg/kg Denguenil, *p* ˂ 0.0001), the necrosis was significantly attenuated to 6% in the 15th day group, indicating 86% inhibition of hepatocyte necrosis compared to Disease Control (15th day, Disease Control, 41% vs. 6%, 140 μg/kg Denguenil, p ˂ 0.0001) (Figure 2E,J and Figure 3A).

### 3.4. Denguenil Inhibited Dengue-Virus-Induced Liver Inflammation

Inflammatory cell accumulation is a hallmark of viral infection. We therefore examined the inflammatory cell accumulation in liver upon dengue viral infection. Normal Control liver smears showed no inflammatory cell accumulation (Figure 3B). However, dengue-virus-induced inflammatory cell accumulation was observed in the livers of Disease Control fish with 19% inflammation on the 8th day; this active inflammation significantly progressed to 22% (p ˂ 0.0001) by the 15th day. The Denguenil treatment groups showed a decrease in liver inflammation compared to Disease Control (Figure 3B). In the 5.6 μg/kg Denguenil group, the inflammation was significantly decreased by 15% in the 8th day group (Disease Control, 19% vs. 16%, 5.6 μg/kg Denguenil, p ˂ 0.0001) and 9% in the 15th day group (Disease Control, 22% vs. 20%, 5.6 μg/kg Denguenil, p ˂ 0.0001) respectively. In the 28 μg/kg Denguenil group, while the inflammation was reduced by 37% in the 8th day group (Disease Control, 19% vs. 12%, 28 μg/kg Denguenil, p ˂ 0.0001), the inflammation was further decreased to 56% in the 15th day group (Disease Control, 22% vs. 9%, 28 μg/kg Denguenil, p ˂ 0.0001) (Figure 3B). Interestingly, 140 μg/kg Denguenil at both time points exhibited significantly abrogated inflammation compared to Disease Control in both the 8th day group (Disease Control, 19% vs. 0%, 140 μg/kg Denguenil, p ˂ 0.0001) and the 15th day group (Disease Control, 22% vs. 0%, 140 μg/kg Denguenil, p ˂ 0.0001) (Figure 3B), suggesting the therapeutic benefit of Denguenil and complete protection at this dose from dengue-virus-induced liver inflammation.

### 3.5. Denguenil Attenuated Dengue-Virus-Induced RBC Infiltration into Liver

We measured virus-induced RBC infiltration in the liver (Figure 3C). The results indicated that the virus induced RBC infiltration in both the 8th day group (Normal Control, 0 vs. 8, Disease Control, *p* ˂ 0.0001) and the 15th day group (Normal Control, 3 vs. 14, Disease Control, *p* ˂ 0.0001) and treatment with Denguenil attenuated vascular leakage and RBC infiltration at both time points. The 140 μg/kg dose of Denguenil showed a maximum attenuation of RBC infiltration compared to Disease Control on both the 8th day (Disease Control, 8 vs. 1, 140 μg/kg Denguenil, *p* ˂ 0.0001) and 15th day (Disease Control, 14 vs. 1, 140 μg/kg Denguenil *p* ˂ 0.0001) (Figure 3C). Further, the maximum percentage of protective response was observed on the 15th day compared to the 8th day (140 μg/kg Denguenil, 8th day, 87% vs. 92%, 15th day, *p* ˂ 0.0001) (Figure 3C).

### 3.6. Dengue-Virus-Infected Zebrafish Blood Phenotyping Identified Protective Effects of Denguenil

Blood phenotyping was performed to assess the accumulation of various inflammatory cells. Normal Control group showed intact and well-defined healthy cells along with very few enlarged and swollen WBCs (Figure 4A(a,f)). While dengue viral infection resulted in increased WBC count, a dose-dependent decrease in WBC was observed at both time points, with the highest percentage of protection in 140 µg/kg Denguenil after 15 days (Figure 4B). Compared to Normal Control, dengue viral infection resulted in accumulation of poorly stained, swollen WBC (Figure 4A(b,g)), in a time-dependent manner (Normal Control, 2.8 × 10^3^ vs. 3.1 × 10^3^, 8th day, Disease Control, *p* ˂ 0.0001; Normal Control, 2.8 × 10^3^ vs. 3.3 × 10^3^, 15th day, Disease Control, *p* ˂ 0.0001). The treatment with 5.6 μg/kg Denguenil on the 8th day did not show protective response and was comparable to Disease Control (Disease Control, 3.1 × 10^3^ vs. 3.1 × 10^3^, 5.6 μg/kg Denguenil) (Figure 4B); Conversely, the WBC count was significantly reduced in the 15th day group upon receiving 5.6 μg/kg Denguenil (Disease Control, 3.1 × 10^3^ vs. 2.9 × 10^3^, 5.6 μg/kg Denguenil, *p* ˂ 0.0001) (Figure 4A(c,h),B). The WBC count was further decreased in the 28 μg/kg Denguenil group on the 15th day compared to the 8th day (28 μg/kg Denguenil, 8th day 3.0 × 10^3^ vs. 2.9 × 10^3^, 15th day, *p* ˂ 0.0001) (Figure 4A(d,i),B). Among the two time points, fish receiving 140 μg/kg Denguenil showed significantly reduced WBC count on the 15th day compared to Disease Control (Disease Control, 3.1 × 10^3^ vs. 2.8 × 10^3^, *p* ˂ 0.0001) (Figure 4A(e,j),B).

Further, RBC counts indicated that viral infection induced a decrease in total RBC count on both the 8th day (Figure 4C) (Normal Control, 3.0 × 10^6^ vs. 2.7 × 10^6^, Disease Control, *p* ˂ 0.0001), and 15th day (Normal Control, 3 × 10^6^ vs. 2.6 × 10^6^, Disease Control, *p* ˂ 0.0001) (Figure 4C); whereas drug treatment induced hemopoiesis in a dose- and time-dependent manner (Figure 4C), with a maximum RBC production after 15 days in fish receiving 140 μg/kg drug (15th day, Disease Control, 2.6 × 10^6^ vs. 3.0 × 10^6^, 140 μg/kg Denguenil, *p* ˂ 0.0001), restoring the RBC number to the Normal Control level (Figure 4C).

Decrease in platelet count is a hallmark of dengue viral infection. From the results, it is evident that dengue virus induced a 50% decrease in platelet count in a time-dependent manner on both the 8th day (Normal Control, 230 × 10^3^ vs., 128 × 10^3^, Disease Control, *p* ˂ 0.0001) and 15th day (Normal Control, 231 × 10^3^ vs. 125 × 10^3^, Disease Control, *p* ˂ 0.0001) (Figure 4D). The platelet numbers were restored upon Denguenil treatment in a dose- and time-dependent manner, with 140 μg/kg Denguenil showing the maximum platelet induction (8th day, Disease Control, 128 × 10^3^ vs. 182 × 10^3^, 140 μg/kg Denguenil, *p* ˂ 0.0001; 15th day, Disease Control, 128 × 10^3^ vs. 191 × 10^3^, 140 μg/kg Denguenil, *p* ˂ 0.0001) (Figure 4D).

In line with blood phenotyping, hematocrit analysis identified a decrease in hematocrit percentage upon dengue viral infection, compared to Normal Control on both the 8th day (Normal Control 33% vs. Disease Control, 31%, *p* ˂ 0.0001) and 15th day (Normal Control 33% vs. Disease Control, 28%, *p* ˂ 0.0001) (Figure 4E). Change in blood cell types upon Denguenil treatment translated as a dose- and time-dependent increase in hematocrit percentage, with the 15th day time point showing the maximum percent of restoration in fish receiving 140 μg/kg Denguenil (Normal Control 33% vs. 33%, 140 μg/kg Denguenil, *p* ˂ 0.0001) (Figure 4E). No significant increase in hematocrit percentage was noticed at the 8th day time point (Disease Control 31% vs. 31%, 140 μg/kg Denguenil, NS) (Figure 4E).

### 3.7. Denguenil Attenuated Dengue-Virus-Induced Hemorrhage

We analyzed caudal fins for signs of virus-induced hemorrhage and presence of blood clots. While clear, unimpaired, and distinct caudal fins were present in normal control (Figure 5A(f)), dengue viral infection resulted in hemorrhage and formation of clot, along with ripped edges of caudal fins, indicating impairment of peripheral cells (Figure 5A(b,g)). In the 8th day group, hemorrhage and well stained thick blood clots were identified in 5.6 μg/kg Denguenil (Figure 5A(c)). At 28 μg/kg Denguenil, palely stained blood clots and hemorrhage with completely fringed caudal tip were present (Figure 5A(d)). At 140 μg/kg Denguenil, partial hemorrhage with small clots and clear blastema cells were observed (Figure 5A(e)). This suggests a partial recovery from hemorrhage at 140 μg/kg dose in 8th day time point. Interestingly in 15th day point, all the three doses (5.6 μg/kg–140 μg/kg) showed substantial recovery from dengue infection induced hemorrhage. In 5.6 μg/kg Denguenil, palely stained hemorrhage clots were noticed (Figure 5A(h)) and at 28 μg/kg Denguenil dose, very few clots were identified with normal looking caudal fin (Figure 5A(i)). At 140 μg/kg dose in 15th day time point, hemorrhage and blood clots were very few with normal, clear and unimpaired caudal fins (Figure 5A(j)). These results indicate that zebrafish receiving Denguenil for 15 days recovered from virus induced hemorrhage, completely as indicated by a dose-dependent decrease in the size of hemorrhage clot (Figure 5B).

### 3.8. Denguenil Normalized Expression of Endothelial Apoptotic Marker, Angiopoietin-2

Next, we tested for the expression of the endothelial cell apoptotic marker Angiopoetin-2 (Ang-2). ANG-2 is an important proangiogenic factor that has recently been implicated in mediating inflammatory processes [37]. Parallel to DENV-3 expression, *Ang-2* expression levels exhibited a 4.6-fold increase on the 8th day (Normal Control, 1.5-fold vs. 7.4-fold, Disease Control, *p* ˂ 0.0001) and 4-fold increase on the 15th day (Normal Control, 2-fold vs. 8-fold, Disease Control, *p* ˂ 0.0001) (Figure 6A). Denguenil treatment resulted in a dose-dependent decrease in *Ang-2* expression levels at both time points tested, with the 140 μg/kg dose exhibiting more than 50% decrease in *Ang-2* expression (8th day group, Disease Control, 7.4-fold vs. 3.3-fold in 140 μg/kg Denguenil, *p* ˂ 0.0001; 15th day group, Disease Control, 8-fold vs. 3-fold in 140 μg/kg Denguenil, *p* ˂ 0.0001) (Figure 6A).

### 3.9. Denguenil Inhibited Upregulation of the CCL3 Gene

Chemokines are involved in leucocyte trafficking during infection and inflammation. Gene expression analysis of *CCL3* indicated a 5-fold increase at the 8th day time point (Normal Control, 30-fold vs. 150-fold, Disease Control, *p* ˂ 0.0001) and 15th day time point (Normal Control, 30-fold vs. 153-fold, Disease Control, *p* ˂ 0.0001) (Figure 6B) upon dengue infection. While a moderate dose-dependent response in the expression of *CCL3* was detected on the 8th day, a robust dose-dependent decrease was identified in the 15th day group upon Denguenil treatment (Figure 6B). In line with this, 140 μg/kg Denguenil in the 8th day group displayed only a 26% decrease in *CCL3* expression compared to Disease Control (Disease Control, 150-fold vs. 110-fold in 140 μg/kg of Denguenil, *p* ˂ 0.0001), whereas a robust 40% decrease in *CCL3* expression was identified in 140 μg/kg of Denguenil after 15 days (Disease Control, 153-fold vs. 91-fold in 140 μg/kg, *p* ˂ 0.0001) (Figure 6B).

### 3.10. Cytosafety of Denguenil in Human Cell Lines

In order to test the cytosafety profile of Denguenil Vati, we first measured cell viability using human hepatic cell line HepG2 and human skin cell line A431. The results indicated that in HepG2 cells, Denguenil did not exert any cytotoxicity up to 300 µg/mL (Figure 7A). In HepG2, very minimal cytotoxicity was observed at 1000 µg/mL. No cytotoxicity was noticed in A431 cells (Figure 7B).

Next, Denguenil-induced oxidative stress was evaluated by measuring the reactive oxygen species (ROS) generation and malondialdehyde (MDA) levels. Treatment with Denguenil did not induce any ROS generation in either HepG2 and A431 cells (Figure 7C,D) even at the highest dose tested. Malondialdehyde (MDA), the major end product of lipid peroxidation and indicator of damage to membrane lipids had no significant changes in either HepG2 or A431 cells even at the highest tested dose of Denguenil (Figure 7E,F). Taken together, Denguenil exhibited an acceptable cytosafety profile in human cell lines under in vitro conditions.

### 3.11. HPLC Analysis of Denguenil

HPLC analysis identified the active ingredients present in the Denguenil. Comparing the chromatograms obtained from the pure substances used to prepare the standard curve identified the presence of gallic acid, ellagic acid, palmetin, and berberine as active ingredients. The analytical curve showed linearity in the concentration range used as standard (data not shown). Gallic acid, ellagic acid, palmetin, and berberine were separated within 40 min. Gallic acid, ellagic acid, palmetin, and berberine were eluted at 7.187 min, 21.240 min, 24.722 min, and 25.036 min retention time, respectively (Figure 8A,B). HPLC analysis revealed that 0.86 μg/mg gallic acid, 1.87 μg/mg ellagic acid, 0.03 μg/mg palmetin, and 0.04 μg/mg berberine were present in Denguenil.

## 4. Discussion

Dengue is an acute febrile illness caused by dengue virus (DENV) and is a major cause of morbidity and mortality in tropical and subtropical regions of the world [2]. However, effective antiviral therapy is lacking. With the absence of an effective anti-dengue viral drug, dengue fever has become a major public health threat worldwide [5]. In the present research, we have successfully demonstrated the therapeutic efficacy and disease-modifying properties of an ayurvedic herbal formulation, Denguenil, on dengue-virus-induced pathology. Further, we have also demonstrated that translational dose of Denguenil is effective in modifying various cellular, molecular, and biochemical diagnostic endpoints of human dengue infection using a physiologically relevant zebrafish model (Table 1).

Zebrafish has been used as model for studying various human diseases and host–pathogen interactions [23]. Further, it has been used to study a number of virus-induced human diseases [22], identify drug leads, and explore new applications for known drugs [38]. In the present investigation, we were successful in exploring the therapeutic benefit of Denguenil against dengue viral infection by using a series of effective doses of Denguenil (5.6–140 μg/kg) more than 1000-fold lower than the human therapeutic dose. Effective dose screening indicated a dose-dependent decrease in various parameters tested, with the maximum response at 140 μg/kg Denguenil after 15 days.

Denguenil is an ayurvedic herbal formulation consisting of extracts from *Tinospora cordifolia* (Giloy or Guduchi), *Aloe vera* (Aloe), *Carica papaya* (Papaya), *Punica granatum* (Pomegranate), *Ocimum sanctum* (Holy Basil). The immunomodulatory property of *Tinospora cordifolia* includes reported anti-human immunodeficiency virus (HIV) activity [31]. In silico analysis identified that it is a potent inhibitor of the NS2B-NS3 receptor in dengue virus [39]. The acemannan carbohydrate polymers present in *Aloe vera* facilitate phagocytosis by acting as a bridge between foreign proteins, such as virus particles and macrophages [40], and reduce hepatic damage by inhibiting necrosis and mononuclear cell infiltration [25,41]. *Carica papaya* leaf extracts exert antidengue properties by decreasing intracellular viral load [42]. Further, it improves platelet and RBC counts [42,43]. However, the exact phytochemicals responsible for boosting thrombopoiesis and erythropoiesis have not been identified. *Punica granatum* seeds contain a high content of punicic acid. It increases peripheral blood mononuclear cell (PBMC) count and inhibits apoptosis in splenocytes and PBMCs [44]. Further, its use as antiviral therapy has been reported against herpes simplex virus (HSV) infection [45] and as an HIV entry inhibitor [46]. *Ocimum sanctum* has been recommended for the treatment of chronic fever, respiratory diseases, and skin diseases. It has antimicrobial, and hepatoprotective properties due to the presence of Eugenol as an active constituent [47], and exerts modest inhibitory activity against DENV-1 [48].

HPLC analysis revealed the presence of gallic acid, ellagic acid, palmetin, and berberine as active chemicals present in Denguenil. Mechanistically, how Denguenil and its active ingredients protect against DENV infection is not very clear at present. However, ellagic acid and gallic acid have been found to inhibit entry of infectious ebolavirus into host cells, and have exhibited potent antifiloviral activity [49]. Further, gallic acid has also been reported to inhibit attachment and entry of human rhinoviruses and HSV type 1 [50,51]. Berberine has been in use for hepatitis [27,52] and as an anti-inflammatory compound [53]. Further, berberine has been shown to inhibit Chikungunya Virus (CHIKV) replication and downregulation of viral protein expression [54]. Palmetin has been found to inhibit respiratory syncytial virus (RSV) replication, along with possessing immunomodulatory properties.

The diagnosis of dengue fever is carried out based on various parameters, including clinical, epidemiological, and laboratory data. Among laboratory tests, the nonspecific tests include blood count, platelet count, prothrombin time (PT), liver function tests, and serum albumin concentration, whereas specific tests include viral isolation tests and serology for antibody examination [55,56]. Keeping these parameters as standard, in a translational approach, we have successfully replicated dengue viral pathology in zebrafish. After the onset of dengue viral illness, the virus can be detected in serum, plasma, circulating blood cells, and other tissues for four to five days. In the present investigation, we used infected serum to propagate the virus in zebrafish and for subsequent studies. Dengue viral infection resulted in a significant increase in total WBC count, whereas the total RBC number was decreased. Hematologically, dengue fever causes early neutropenia, with subsequent lymphocytosis and a decrease in platelet count and hypocellular marrow with abnormal megakaryocytopoiesis in the early stage [57]. In the present investigation, increased WBC number could be due to the presence of active and ongoing dengue viral infection. Decreased RBC number upon infection is indicative of poor hemopoiesis and vascular leakage induced upon viral infection. Conversely, increased platelet counts in zebrafish receiving drug suggests the restoration of hemopoiesis upon Denguenil treatment.

Upon infection, the dengue virus can infect various cell types and cause diverse clinical and pathological effects. Histopathology suggests that various organs, including the lymph nodes, spleen, liver, and bone marrow, are affected by dengue viral replication, and a variety of cell types, including macrophages, lymphocytes, dendritic cells (DCs), monocytes, and endothelial cells become infected at varying rates during the course of disease [58]. In the present study, we have observed a severe liver pathology in the zebrafish model, including hepatic cell necrosis and inflammation. The observed liver damage in our model could be due to dysregulated host immune response and its products (e.g., cytokines and chemokines) or the direct dengue viral infection of the cells [59]. Previous reports suggest that liver damage in dengue includes microvascular steatosis, hepatocellular necrosis, and cellular infiltrates at the portal tract [11]. In the present investigation, we have demonstrated that therapeutic administration of Denguenil has protective effects on virus-induced liver damage as evidenced by reduced necrosis and inflammation in a dose- and time-dependent manner. Further, liver pathological symptoms were abrogated by Denguenil. This indicates that the drug itself does not induce inflammatory responses and has no toxic effect on liver, as shown by its acceptable cytosafety profile in human cells. Involvement of JNK1/2 signaling, ERK1/2 pathway, and p38 MAPK signaling pathways in liver damage were reported, and use of their inhibitors reduced hepatic cell apoptosis and liver injury in DENV-infected mice; however, these inhibitors were not able to restrict virus replication in the liver [60,61,62].

The dengue virus can replicate in both hepatocytes and Kupffer cells [63]. In the present study, we have observed an increase in the transcript copies of DENV-3 by 40-fold upon infection with serum-containing active viruses. The copy number did not change after the 8th day until the end of the experiment in the Disease Control. Significantly increased copy number is indicative of active viral replication, multiplication, and propagation of disease. Use of various replication and transcription inhibitors, such as nucleoside analogs [64,65], helicase [66], and protease inhibitors [67] have been tested as potential dengue therapy. However, they either failed in in vivo models or human trials, suggesting their development is still in infancy [26]. A significant attenuation of viral copy number upon Denguenil treatment indicates its efficacy as a therapeutic agent.

Thrombocytopenia is one of the simple diagnostic criteria proposed by the World Health Organization (WHO) for diagnosis of dengue hemorrhagic fever (DHF) [68]. Dengue complications are frequently preceded by a rapid drop in platelet counts, often to very low numbers [69]. Decrease in platelet count in the zebrafish model is in line with the severe dengue fever symptomology in humans. Further dose-dependent increase in platelet count in effective dose screening at both 8th and 15th day time points indicates the superior role of Denguenil on DENV-induced thrombocytopenia. Interestingly, *Carica papaya* leaf extract has been found to be an antidengue agent, and resulted in platelet augmentation and significant decrease in erythrocyte damage [42]. In the present study, while we have demonstrated the restoration of platelet count upon Denguenil treatment, the mechanism leading to protective response from thrombocytopenia is yet to be ascertained.

Vascular leakage, which is a hallmark of DHF, has been found to be associated with derangement of the normal regulatory function of endothelial cells, rather than endothelial cell death, and underlies the hemorrhagic diathesis of dengue [70]. Numerous studies have established ANG-2 and CCL3 as biomarkers that serve as reference standards for sensitive and early indicators of disease [14,71,72,73]. Hence, even with minute onset of viral replication and pathology, these genes show changes in their steady-state mRNA levels. In the present study, we have used these two genes to establish the fact that abrogation of ANG-2 and CCL3 mRNA levels to baseline is a strong indicator of drug efficacy. This also indicates the pathways that may be inhibited at the early stage of the disease. In the present research, the observed decrease in RBC count could be due to vascular leakage, and coincides with the expression of apoptotic marker *ANG2*. Previous reports suggest that ANG2 is pro-apoptotic and that its expression is associated with endothelial breakdown and apoptosis of vascular cells during injury [74]. Interestingly, significant attenuation of *ANG2* expression and restoration of RBC number upon Denguenil treatment is indicative of anti-apoptotic and anti-dyserythropoietic properties of Denguenil. In line with WBC, RBC, and platelet counts, dengue virus induced a decrease in hematocrit percentage, which was restored upon drug Denguenil treatment. Dose-dependent decrease in dengue parameters without any overt pathological features further indicates the beneficial effect of this ayurvedic herbal formulation.

In the present study, the clinical course of hemorrhage was established as per the WHO (2009) [9], CDC, and previous reports [75]. We observed hemorrhage during the initial phase of the study, i.e., from day zero to seven. This is due to the leaky vessels at the initial stage of the infection with visible RBC. After seven days, the hemorrhage stopped and the Disease Control survived. However, blood clotting was still observed in the periphery of the caudal fin with indicative darkly stained blotches and uneven fringes in the caudal fin tip clearly visible. Treatment with Denguenil resulted in recovery of DC from dengue-virus-induced hemorrhage as evident by minimal blotches and decreased peripheral vascular damage.

Previous reports indicate that CCL3 levels are increased in liver and spleen of DENV-infected mice [76] and elevated CCL3 concentrations in plasma of patients with DENV infection may be associated with disease severity [77]. Further, reports suggest that CCL2 and CCL3 may play a role in early recruitment of leukocyte [78] subsets and may have a strong influence in patient clinical outcome [79].

Use of natural compounds have been found to be effective in inhibiting DENV infection and have the potential to be developed as anti-dengue drugs [32,80]. However, many natural products that have shown potent inhibitory activity against certain viruses in vitro have failed to progress to clinical trials because of their poor in vivo activity [81]. In addition, several natural products, such as Quercetin and Narasin, have also been observed to possess significant anti-DENV properties [82,83].

## 5. Conclusions

In the present study, we successfully developed a novel zebrafish model of dengue virus pathology and demonstrated the therapeutic efficacy and disease-modifying properties of the Indian traditional pentaherbal formulation, Denguenil Vati, on dengue-virus-induced pathology using various cellular, molecular, and biochemical diagnostic endpoints. Our zebrafish model of dengue viral pathology would help in early screening of potential anti-dengue drugs and aid in proof-of-concept studies, thereby reducing the gap in drug development and increasing overall drug success rates. Use of Denguenil Vati formulation as a complementary or alternative medicine in dengue treatment could alleviate dengue pathological features. The present preclinical study warrants clinical trials of Denguenil Vati in human subjects of dengue and other viral diseases.

## Figures and Tables

**Figure 1 biomolecules-10-00971-f001:**
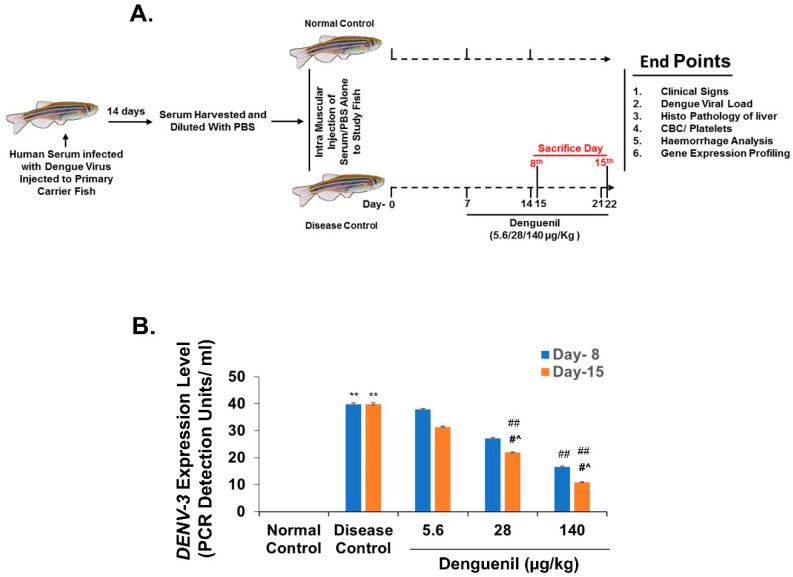
Denguenil inhibits dengue virus (DENV) transcript levels. (**A**) Schematic representation of zebrafish model of dengue viral pathology. Human serum infected with dengue virus was propagated in primary carrier fish. Study zebrafish (n = 24) were injected with dengue virus (DENV-3) (Disease Control) or PBS as in control (Normal Control) and treated with three different effective doses (5.6 μg/kg, 28 μg/kg, and 140 μg/kg Denguenil) for various time points. (**B**) Whole RNA from liver tissue was analyzed by quantitative real-time PCR (qPCR) for the dengue viral load. Virus induced increases in the DENV-3 copy number were decreased upon Denguenil treatment. n = 24, data are presented as means ± SEM. **, ##, #^ *p* < 0.001, * Represents significant compared to Normal Control; ##, # Represents significant compared to their respective Disease Controls; #^ Represents significant among two time points.

**Figure 2 biomolecules-10-00971-f002:**
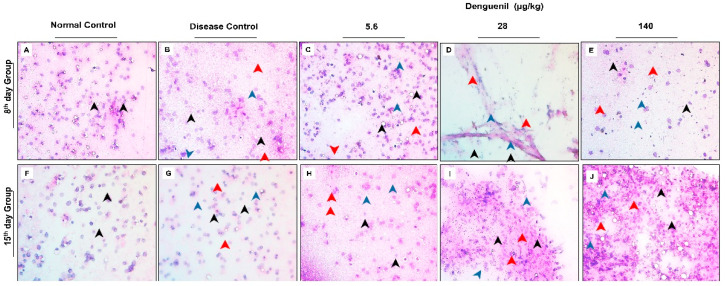
Smear pathology examination of liver showing the protective effect of Denguenil. Hematoxylin and eosin (H&E) staining and analysis was performed on liver smears from different groups containing various doses to assess the anatomy and cellular damage upon DENV-3 infection, and protective effect of Denguenil on cellular phenotypes. (**A**, **F**) Normal Control. (**B**, **G**) Disease Control. (**C**, **H**) 5.6 μg/kg Denguenil. (**D**, **I**) 28 μg/kg Denguenil. (**E**) 140 μg/kg Denguenil, 8th day group. (**J**) 140 μg/kg Denguenil, 15th day group. Black arrowheads indicate normal and intact cells; red arrowheads indicate necrotic cells, and blue arrowheads indicate inflammatory cells. Representative images from each group (n = 24) are shown. Original magnification, 40×.

**Figure 3 biomolecules-10-00971-f003:**
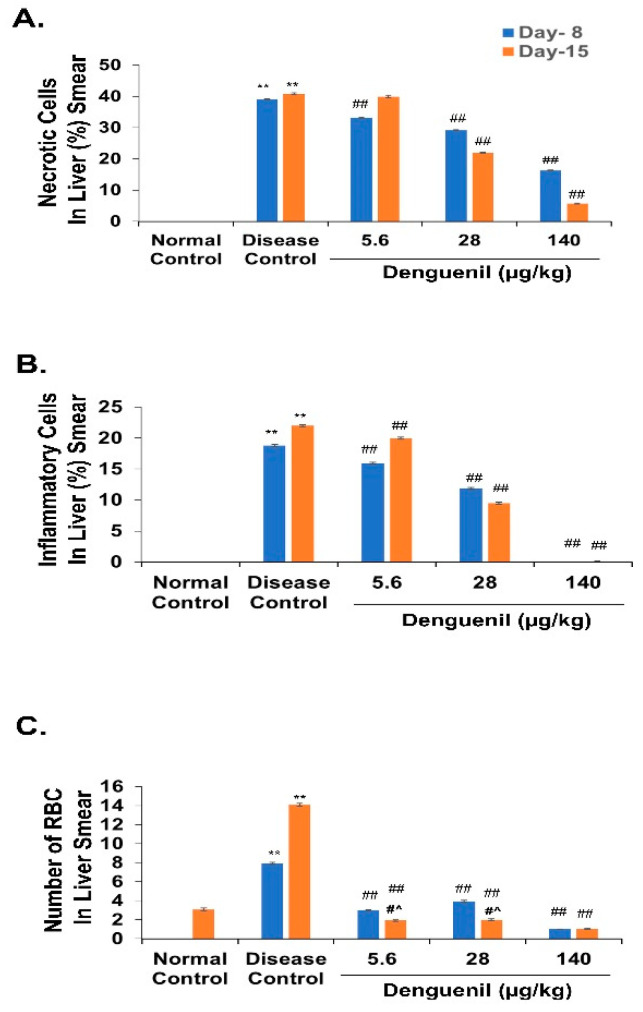
Denguenil treatment inhibits hepatic cell necrosis and inflammation, and red blood cell (RBC) infiltration into liver. H&E staining and analysis was performed on liver smears from different groups at the 8th day and 15th day time points. Percentage necrosis, inflammation, and RBCs present in liver smear were quantified. (**A**) Denguenil treatment inhibited the dengue-virus-induced hepatic cell necrosis in a dose-dependent manner with the maximum protection at 15th day time point in the 140 μg/kg group. (**B**) Denguenil treatment inhibited the dengue virus induced liver inflammation in a dose-dependent manner. (**C**) RBC infiltration into liver was inhibited upon Denguenil treatment. n = 24, data are presented as means ± SEM. **, ##, #^ *p* < 0.001. ** Represents significant compared to Normal Control; ## Represents significant compared to their respective Disease Controls; #^ Represents significant among two time points.

**Figure 4 biomolecules-10-00971-f004:**
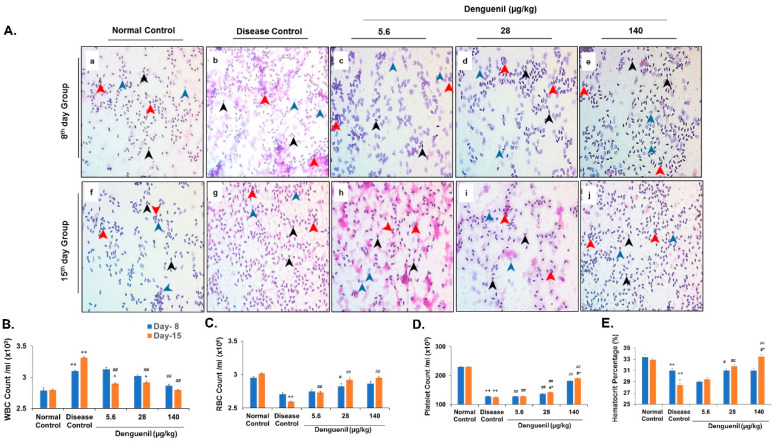
Pharmacological treatment with Denguenil reverses virus-induced serological dynamics. Whole blood was collected and analyzed using H&E staining of blood smears and differential blood cell counts. (**4A**) Blood smear analysis. (**a**, **f**) Normal Control. (**b**, **g**) Disease Control. (**c**, **h**) 5.6 μg/kg Denguenil. (**d**, **i**) 28 μg/kg Denguenil. (**e**, **j**) 140 μg/kg Denguenil. (**4B**) Whole blood WBC count. (**4C**) Whole blood RBC count. (**4D**) Whole blood platelet count. (**4E**) Hematocrit percentage analysis. Black arrowheads indicate RBC; blue arrowheads indicate WBC, and red arrowhead indicates platelets. n = 24, data are presented as means ± SEM. **, ##, #^ *p* < 0.001 and *, #, ^ *p* < 0.05 by one-way ANOVA. **, * Represents significant compared to Normal Control; ##, #Represents significant compared to their respective Disease Controls; #^, ^Represents significant among two time points. Original magnification, 40×.

**Figure 5 biomolecules-10-00971-f005:**
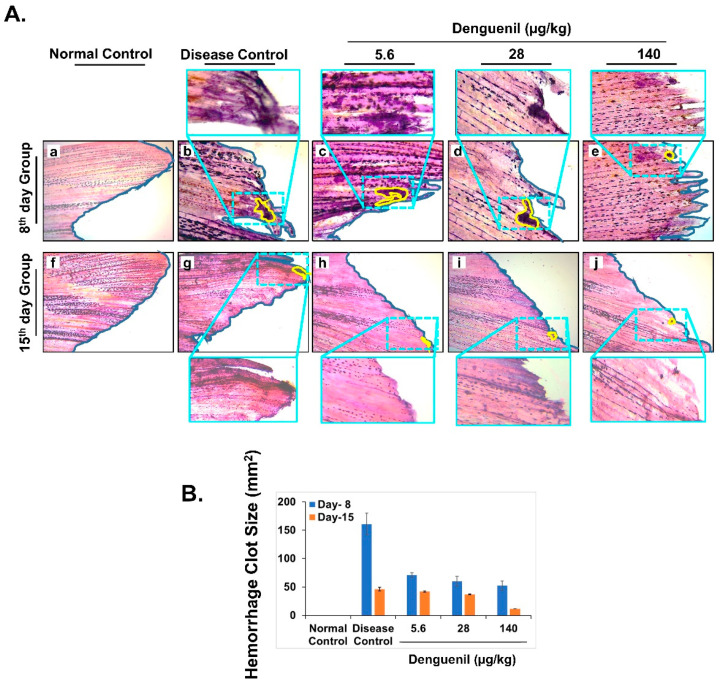
Pharmacological treatment with Denguenil inhibits dengue-virus-induced hemorrhage. Zebrafish caudal fins were harvested from all the groups at the end of the experiment and analyzed for dengue-virus-induced hemorrhage, at the 8th day and 15th day time points. (**A**) H&E stained caudal fin anatomy. 5A. (**a**, **f**) Normal Control. 5A. (**b**, **g**) Disease Control. 5A. (**c**, **h**) 5.6 μg/kg Denguenil. 5A. (**d**, **i**) 28 μg/kg Denguenil. 5A. (**e**, **j**) 140 μg/kg Denguenil. The Caudal fin areas with observed hemorrhage and clots (marked in yellow) have been zoomed expanded for clarity. Original magnification, 10×, zoomed in magnification, 40×. (**B**) Quantification of hemorrhage clot size, averaged across each treatment group and respective time points.

**Figure 6 biomolecules-10-00971-f006:**
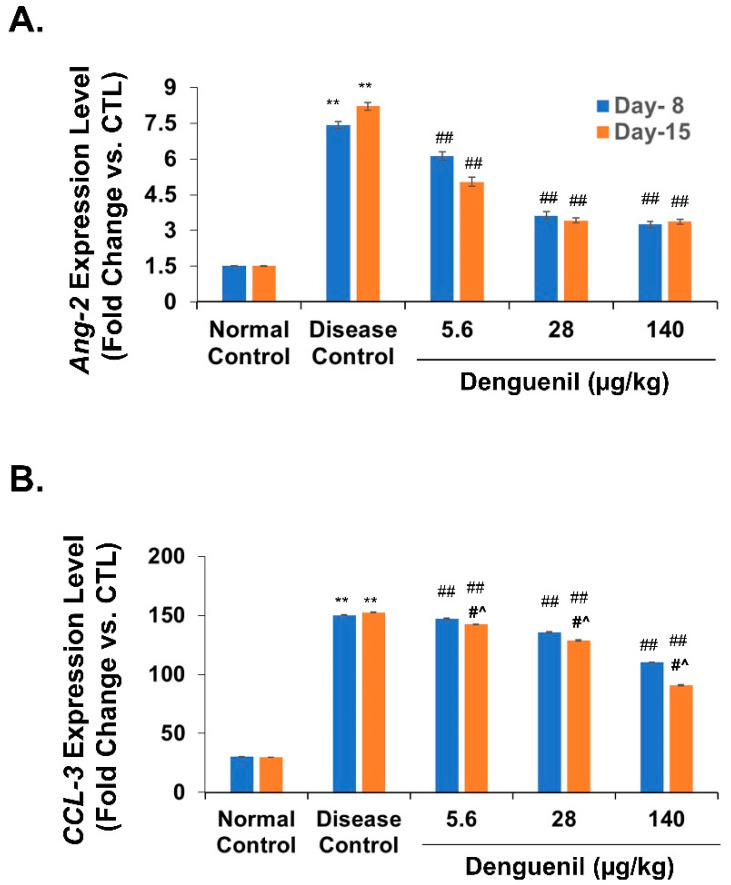
Denguenil treatment normalizes expression of *Ang2* and chemokine *CCL3*. Virus-induced increases in (**A**) *Ang-2* expression and (**B**) *CCL3* levels were attenuated upon Denguenil treatment in a dose-dependent manner. n = 24, data are presented as means ± SEM. **, ##, #^ *p* < 0.001. ** Represents significant compared to Normal Control; ##, Represents significant compared to their respective Disease Controls; #^ Represents significant difference between the two time points.

**Figure 7 biomolecules-10-00971-f007:**
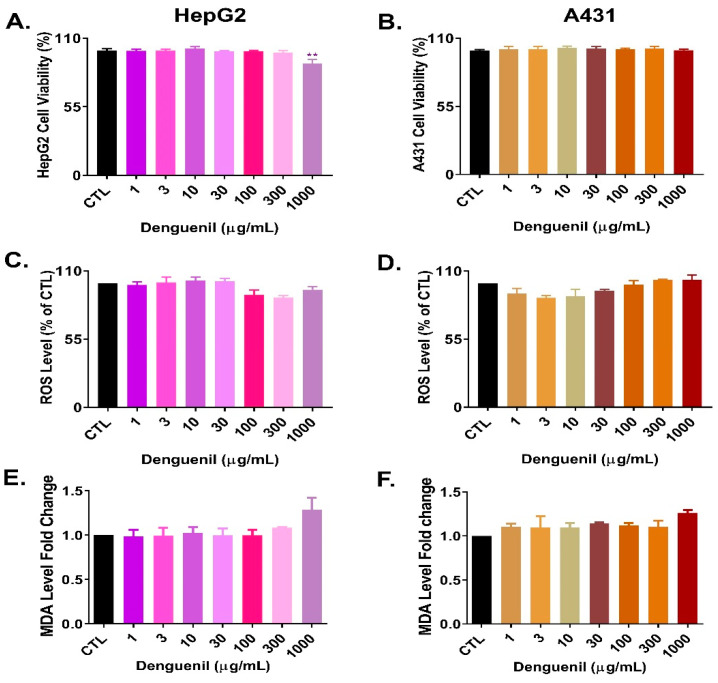
Cytosafety of Denguenil in human cell lines. HepG2 and A431 cells were treated with a series of Denguenil concentrations for 24 h and tested for various endpoints. In (**A**) HepG2 cells and in (**B**) A431 cells, Denguenil exerted no cytotoxicity up to the highest tested concentration of 1000 µg/mL. Reactive oxygen species (ROS) generation was measured (**C**) In HepG2 and (**D**) A431 cells. MDA secretion in cell culture supernatant was estimated in (**E**) HepG2 and (**F**) A431 cells. n = 3, data are presented as means ± SEM. **, *p* < 0.001, represents statistically significant as compared to untreated control (CTL).

**Figure 8 biomolecules-10-00971-f008:**
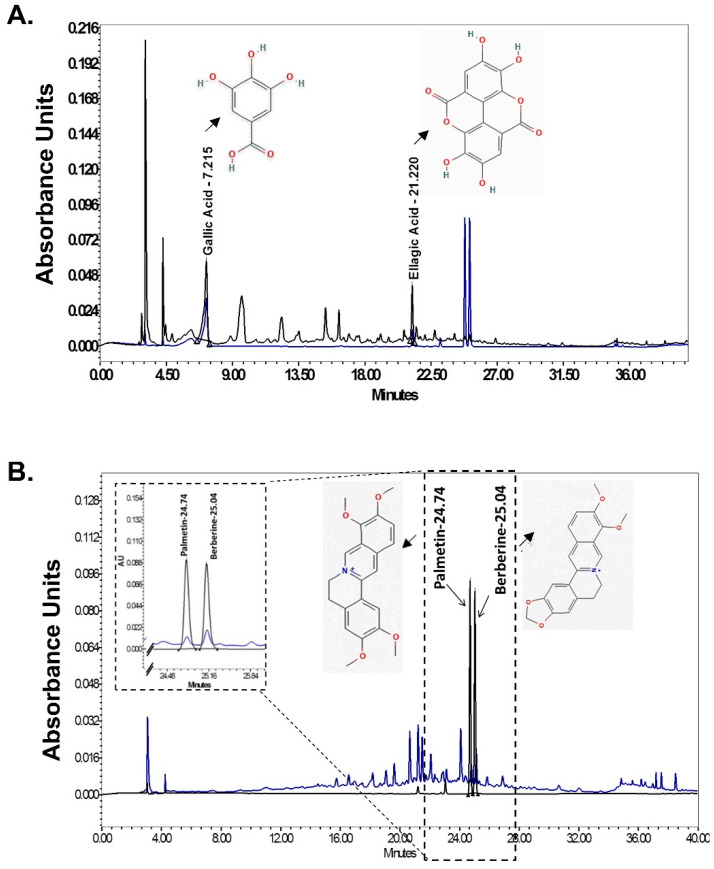
HPLC analysis identification of active ingredients in Denguenil. The Denguenil sample was crushed into a fine powder and analyzed on HPLC using reference standards. The chromatographs were recorded at 270 nm (gallic acid and ellagic acid) and 346 nm (palmetin and berberine) wavelength. By comparing with chromatographs of pure standards, HPLC analysis identified the presence of bioactive compounds, namely, (**A**) gallic acid and ellagic acid and (**B**) palmetin and berberine in the Denguenil. The chemical structures of the identified compounds have been appended in the chromatogram.

**Table 1 biomolecules-10-00971-t001:** The course of dengue infection and pathological changes. Various pathological features of dengue infection in zebrafish, its comparison with human dengue clinical signs, and subsequent therapeutic modulation by Denguenil treatment.

Course of Dengue Illness	Zebrafish Aetiology			Human Dengue Endpoints
	Disease Initiation	Critical Progression	Recovery by Denguenil Vati	
**Phenotypic Changes**	Hypoactivity and Body discolouration	Vascular Hemorrhage of blood vessels in the caudal extremities	Caudal tip with no fringes	Lethargy, Restlessness, Endothelial dysfunction, Vascular leak and Hemorrhage development
**Pathological Changes**	RBC infiltration	Hepatocyte necrosis	Normal hepatocytes and Reduction in RBC infiltration	Hepatocellular necrosis and Cellular infiltration
**Molecular Changes**		Dengue biomarker *DENV-3* expression	No expression	*DENV* transcript expression
		Expression of *Ang-2*	Low expression of *Ang-2*	Elevated *Ang-2* expression
		Increased *CCL3* expression	Low *CCL3* expression	Elevated *CCL3* levels associated with disease severity
**Blood Counts**		Decline in RBC count Increase in WBC count Decline in Platelet count	Restoration of Blood count for RBC- 3 × 10^6^ WBC- 2.8 × 10^3^ Platelet- 191 × 10^3^	Anemia Elevated lymphocytes Decrease in platelet counts

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
