# Peer review of "Validation of a Novel Zebrafish Model of Dengue Virus (DENV-3) Pathology Using the Pentaherbal Medicine Denguenil Vati"

_biomolecules, 2020, doi:10.3390/biom10070971_

Round 1

Reviewer 1 Report

As noted in the review of the original manuscript, this work revealed by using a zebrafish model that Denguenil, penta-herbal medicine, may possess potential therapeutic properties to against serotype 3 of dengue virus (DENV-3) infection. Zebrafishes were injected with viruses propagated in carrier fishes that were infected with serum isolated from patients infected with DENV-3 and treated with three different amounts of Denguenil for 8 and 15 days. The qPCR showed that Denguenil not only reduced viral RNA copy numbers but also inhibit DENV-3-induced hepatocytic necrosis, liver inflammation, RBC infiltration in liver and haemorrhage. Furthermore, Denguenil suppressed DENV-3-induced the expression of Ang-2 and CCL-3, both are apoptotic marker and chemokine gene, respectively. However, some experiments seem to be preliminary and there are several review points listed below that need to be addressed or improved:

Review points:

  1. In addition to TCID50, the titer of DENV-3 may be determined by using plaque assay.
  2. The clinical information of patients who provided serum is not clear.
  3. In Figure 1, why did DENV-3 RNA not elevate at Day-15, compared to one at Day-8?
  4. The morphology of cells in figure 2 and 4 are not clear. The magnification of image should be enhanced. The scale bar should be indicated.
  5. In figure 3A, the author did not explain why there was no time-dependent increasing for necrosis cells in the disease control group.
  6. In figure 5, why did the size of hemorrhage clot decrease at Day-15 in the disease control group?
  7. In figure 6, there is no indication why the author examine ANG-2 and CCL3 particularly and the author should provide evidence in protein level of these two genes.
  8. In figure 7, why the author examine specifically the cyto-satety in HepG2 and A431 cells?
  9. The author should provide evidence to demonstrate ingredients indeed possess anti-DENV-3 activities.
  10. Does Denguenil present anti-viral activity for other types of DENV?
  11. In line 260, there is no Fig 1C in manuscript.

Author Response

Comments and Suggestions for Authors

As noted in the review of the original manuscript, this work revealed by using a zebrafish model that Denguenil, penta-herbal medicine, may possess potential therapeutic properties to against serotype 3 of dengue virus (DENV-3) infection. Zebrafishes were injected with viruses propagated in carrier fishes that were infected with serum isolated from patients infected with DENV-3 and treated with three different amounts of Denguenil for 8 and 15 days. The qPCR showed that Denguenil not only reduced viral RNA copy numbers but also inhibit DENV-3-induced hepatocytic necrosis, liver inflammation, RBC infiltration in liver and haemorrhage. Furthermore, Denguenil suppressed DENV-3-induced the expression of Ang-2 and CCL-3, both are apoptotic marker and chemokine gene, respectively. However, some experiments seem to be preliminary and there are several review points listed below that need to be addressed or improved:

Review points:

  1. In addition to TCID50, the titer of DENV-3 may be determined by using plaque assay.

Response: We thank the reviewer for the logical suggestion. In the present investigation, we employed the only TCID50 to identify the best working dose for the study. While Plaque assay is robust; translating from a invitro to in vivo would require additional standardization steps. Similarly, previous reports where successful experiments which involved the infection of zebrafish with viruses such as Chikungunya, Influenza A and viral encephalitis, the TCID50 method had been employed to generate reproducible models with measurable outcome [1][2][3]. Hence the TCID50 was preferred for this study.

  1. The clinical information of patients who provided serum is not clear.

Response: We thank the reviewer for the valuable point, and our apologies for this hindsight. The samples were obtained from a total of 3 dengue positive male subjects, within age groups of 21 to 45 who had fever accompanied with joint and muscle pain. All the patients confirmed to not having other comorbid conditions. This information has now been captured in section 2.3 of methods.

  1. In Figure 1, why did DENV-3 RNA not elevate at Day-15, compared to one at Day-8?

Response: We thank the reviewer for the logical query. In the present study, the levels of DENV-3 RNA were not elevated after in day-15 group, which was comparable to day-8 group. This is indicative of critical phase of dengue infection in humans. In the critical phase (i.e, – day 8) there is an increased viral replicative cycle and associated pathology. Considering that by day 15 the pathology has diminished with a lesser viral load is indicative of recovery phase that the study group goes through [4][5].

  1. The morphology of cells in figure 2 and 4 are not clear. The magnification of image should be enhanced. The scale bar should be indicated.

Response: We thank the reviewer for the valuable suggestion. As suggested we now enlarged the magnification of the figures 2 and 4. This was presented in the revised figures 2 and 4.  Regarding the smears, considering the study groups are infected and the tissues is soft, sensitive, has a poor stain uptake character [6]. We have retained the quality at which all images are comparable without addition of artificial colours or cartooning effects.

  1. In figure 3A, the author did not explain why there was no time-dependent increasing for necrosis cells in the disease control group.

Response: We thank the reviewer for the logical query. The necrotic patterns in zebrafish is very sensitive, if you notice that zebrafish is an active animal and constantly swim about to and fro except for bouts of sleep. Now any localized inflammation and apoptosis leads to lack of availability of oxygen, this is also associated with gill and fin infections, eventually followed by death. This is generally observed as skin discolouration, poor swim pattern and reduced food intake. In the case of necrosis this entire pathway is accelerated. For instance, a necrotic pattern in the caudal fin or swim bladder will render the fish motionless and leads to death within days if not within one day. Hence necrosis has been tightly monitored in the zebrafish model induction to levels where there is no mass mortality. In the below reference papers [7][8][9], you would notice that from the onset of necrosis to organ pathology is within few hours and groups measured for necrosis were maintained at levels less than 15%, which is consistent with our findings as well.

  1. In figure 5, why did the size of hemorrhage clot decrease at Day-15 in the disease control group?

Response: We thank the reviewer for the valuable query. Several studies have established that coagulopathies drop with time [10][11]. Haemorrhage is observed only during the early virus replication cycle where there is plasma leakage followed by platelet activation, which is highest between day 4 to day 8 due to active viral replication. Now considering the query on the RNA levels on day 15 this is in coherence with the haemorrhage clot observed in this study (elevated on day 8 followed by drop by day 15) which is indicative of the critical phase of dengue infection in humans. In the critical phase there is an increased viral replicative cycle and associated pathology at day-8. Considering that by day-15 the pathology has diminished with a lesser viral load is indicative of recovery phase that the study group goes through.

  1. In figure 6, there is no indication why the author examine ANG-2 and CCL3 particularly and the author should provide evidence in protein level of these two genes.

Response: We thank the reviewer for the important point. Previous reports indicate that ANG2 is pro-apoptotic and its expression is associate with endothelial breakdown and apoptosis of vascular cells during injury [12]. In dengue infection, vascular leakage due to derangement of the normal regulatory function of endothelial cells leads to hemorrhagic diathesis of dengue [13]. Previous reports indicate that CCL3 levels are increased in liver and spleen of DENV infected mice [14] and elevated CCL3 concentrations in plasma of patients with DENV infection may be associated with disease severity [15]. Further, reports suggest that CCL2 and CCL3 may play a role in early recruitment of leukocytes [16] subsets and may have a strong influence in the clinical outcome of patient [17].

Numerous studies have established that both ANG-2 and CCL3 as biomarkers and serves as reference standards for sensitive and early indicators of disease [18][19][20][21]. Hence, even with a minute onset of viral replication and pathology these genes show changes in their steady state mRNA levels. In the present study we have used these two genes to establishing the fact that abrogation of ANG-2 and CCL3 mRNA levels to baseline would be a strong indicator of the drug efficacy.  Further, this will also indicate the pathways that may be inhibited at the early initiation of the disease.

It would be interesting to look at the protein levels of ANG-2 and CCL3. However, due to limited resources and reagent unavailability at the time of study, we could not continue the study beyond mRNA expression levels. We thank the reviewer for the valuable insight. We would definitely plan around this suggestion for the future studies.

Further, in the present investigation, the additional scope was to evaluate the point of disease progress which was pre cascade hence mRNA levels were preferred over protein expression.  Mechanistically, ANG-2 and CCL3 plays a central role in plasma leakage and endothelial activation. Once the pathology proceeds beyond ANG-2 activation to a state of platelet leakage, the cascade burst to a platelet monocyte interaction leading to cytokine storm which immediately leads to glycocalyx degradation and coagulative pathology. Hence the assessment of mRNA at this stage could provide a clear indication on the state of disease progression than the stage of protein expression at detectable levels for ANG-2 and CCL3. Such information has now been captured in the paragraph 9 of revised discussion part:

  1. In figure 7, why the author examine specifically the cyto-satety in HepG2 and A431 cells?

Response: We thank the reviewer for the logical query. In order to test the cyto-safety profile of Denguenil Vati, we first measured cell viability using human hepatic cell line HepG2. We used HepG2 due to the fact that the most drugs first-pass through liver and undergo drug metabolism to produce active metabolites or inactive drugs via the activity of cytochrome P-450 enzymes.

Further, upon infection, the dengue virus can infect various cell types and organs and liver is the commonest organ at varying rates during the course of disease and cause diverse clinical and pathological effects [22] [23]. Indeed, dengue has been implicated as an important cause of acute liver failure in endemic countries [24]. Hence we tested the cyto safety of Denguenil using HepG2.

Conversely, we have also tested for the cytosafety of Denguenil using non-liver cell line, A431, the human skin cell line. Dengue virus is delivered to humans via mosquito bites to the skin and it has been suggested to be a potential target for dengue virus infection [25] [26]. Further, in the present investigation, we have noticed caudal fin haemorrhage and caudal fins cells are analogous to human skin cells, we explored the cyto safety of denguenil using human skin cell line, A431.

  1. The author should provide evidence to demonstrate ingredients indeed possess anti-DENV-3 activities.

Response: Herbal formulation, Denguenil consisting of extracts from Tinospora cordifolia (Giloy), Aloe vera (Aloe), Carica papaya (Papaya), Punica granatum (Pomegranate), Ocimum sanctum (Holy Basil).

The immunomodulatory property of Tinospora cordifolia includes reported anti-HIV activity [27]. In silico analysis identified that it is a potent inhibitor of NS2B-NS3 receptor in Dengue virus [28]. The acemannan carbohydrate polymers present in Aloe vera facilitate phagocytosis by acting as a bridge between foreign proteins like virus particles macrophages [29]. Carica papaya leaf extracts exert anti-dengue properties by decreasing intracellular viral load, confirming its antiviral activity [30]. Punica granatum used as antiviral therapy against HSV infection [31] and also as HIV entry inhibitor [32]. Ocimum sanctum extract exerted modest inhibitory activity against DENV-1 [33]. This information was already captured in the 3rd paragraph of revised discussion.

  1. Does Denguenil present anti-viral activity for other types of DENV?

Response: We thank you for the valuable futuristic query. At present we assessed efficacy of denguenil against DENV-3 as it is a prevalent virus type in the Indian demography. At present, we have not tested denguenil against other types of Dengue. However, we are in the process of designing experiments in such direction. Thank you very much indeed.

  1. In line 260, there is no Fig 1C in manuscript.

Response: We thank the reviewer for the careful reading of the manuscript. We apologise for the typographical error. In the revised manuscript we have edited the line 260 and changed Fig 1C to Fig 1B.

REFERENCES

  1. Palha, N.; Guivel-Benhassine, F.; Briolat, V.; Lutfalla, G.; Sourisseau, M.; Ellett, F.; Wang, C.H.; Lieschke, G.J.; Herbomel, P.; Schwartz, O.; et al. Real-Time Whole-Body Visualization of Chikungunya Virus Infection and Host Interferon Response in Zebrafish. PLoS Pathog. 2013, doi:10.1371/journal.ppat.1003619.
  2. Gabor, K.A.; Goody, M.F.; Mowel, W.K.; Breitbach, M.E.; Gratacap, R.L.; Witten, P.E.; Kim, C.H. Influenza A virus infection in zebrafish recapitulates mammalian infection and sensitivity to anti-influenza drug treatment. DMM Dis. Model. Mech. 2014, doi:10.1242/dmm.014746.
  3. Passoni, G. Unraveling viral encephalitis in vivo : dynamic imaging of neuro-invasion and neuro inflammation processes in the zebrafish, 2015.
  4. Patterson, J.; Sammon, M.; Garg, M. Dengue, zika and chikungunya: Emerging arboviruses in the new world. West. J. Emerg. Med. 2016.
  5. Assir, M.K.Z, Guidelines for clinical case management of dengue fever/ dengue hemorrhagic fever/ dengue shock syndrome 2011 in pakistan context; 2011.
  6. Conrad, R.; Castelino-Prabhu, S.; Cobb, C.; Raza, A. Cytopathologic diagnosis of liver mass lesions. J. Gastrointest. Oncol. 2013.
  7. Roca, F.J.; Whitworth, L.J.; Redmond, S.; Jones, A.A.; Ramakrishnan, L. TNF Induces Pathogenic Programmed Macrophage Necrosis in Tuberculosis through a Mitochondrial-Lysosomal-Endoplasmic Reticulum Circuit. Cell 2019, doi:10.1016/j.cell.2019.08.004.
  8. Xu, X.; Zhang, L.; Weng, S.; Huang, Z.; Lu, J.; Lan, D.; Zhong, X.; Yu, X.; Xu, A.; He, J. A zebrafish (Danio rerio) model of infectious spleen and kidney necrosis virus (ISKNV) infection. Virology 2008, doi:10.1016/j.virol.2007.12.026.
  9. Binesh, C.P. Mortality due to viral nervous necrosis in zebrafish Danio rerio and goldfish Carassius auratus. Dis. Aquat. Organ. 2013, doi:10.3354/dao02605.
  10. Wills, B.A.; Oragui, E.E.; Stephens, A.C.; Daramola, O.A.; Dung, N.M.; Loan, H.T.; Chau, N.V.; Chambers, M.; Stepniewska, K.; Farrar, J.J.; et al. Coagulation Abnormalities in Dengue Hemorrhagic Fever: Serial Investigations in 167 Vietnamese Children with Dengue Shock Syndrome. Clin. Infect. Dis. 2002, doi:10.1086/341410.
  11. Ojha, A.; Nandi, D.; Batra, H.; Singhal, R.; Annarapu, G.K.; Bhattacharyya, S.; Seth, T.; Dar, L.; Medigeshi, G.R.; Vrati, S.; et al. Platelet activation determines the severity of thrombocytopenia in dengue infection. Sci. Rep. 2017, doi:10.1038/srep41697.
  12. Nag, S.; Papneja, T.; Venugopalan, R.; Stewart, D.J. Increased angiopoietin2 expression is associated with endothelial apoptosis and blood-brain barrier breakdown. Lab. Investig. 2005, 85, 1189–1198, doi:10.1038/labinvest.3700325.
  13. Dewi, B.E.; Takasaki, T.; Kurane, I. In vitro assessment of human endothelial cell permeability: Effects of inflammatory cytokines and dengue virus infection. J. Virol. Methods 2004, 121, 171–180, doi:10.1016/j.jviromet.2004.06.013.
  14. Guabiraba, R.; Marques, R.E.; Besnard, A.G.; Fagundes, C.T.; Souza, D.G.; Ryffel, B.; Teixeira, M.M. Role of the Chemokine Receptors CCR1, CCR2 and CCR4 in the Pathogenesis of Experimental Dengue Infection in Mice. PLoS One 2010, 5, e15680, doi:10.1371/journal.pone.0015680.
  15. Spain-Santana, T. a; Marglin, S.; Ennis, F. a; Rothman, a L. MIP-1 Alpha and MIP-1 Beta Induction by Dengue Virus. J. Med. Virol. 2001, doi:10.1002/jmv.2037 [pii].
  16. Rothman, A.L. Immunology and Immunopathogenesis of Dengue Disease. Adv. Virus Res. 2003, 60, 397–419, doi:10.1016/S0065-3527(03)60010-2.
  17. Moreno-Altamirano, M.M.B.; Romano, M.; Legorreta-Herrera, M.; Sánchez-García, F.J.; Colston, M.J. Gene expression in human macrophages infected with dengue virus serotype-2. Scand. J. Immunol. 2004, 60, 631–638, doi:10.1111/j.0300-9475.2004.01519.x.
  18. Mapalagamage, M.; Handunnetti, S.M.; Wickremasinghe, A.R.; Premawansa, G.; Thillainathan, S.; Fernando, T.; Kanapathippillai, K.; de Silva, A.D.; Premawansa, S. High levels of serum angiopoietin 2 and angiopoietin 2/1 ratio at the critical stage of dengue hemorrhagic fever in patients and association with clinical and biochemical parameters. J. Clin. Microbiol. 2020, doi:10.1128/JCM.00436-19.
  19. Malavige, G.N.; Ogg, G.S. Pathogenesis of vascular leak in dengue virus infection. Immunology 2017.
  20. Tolfvenstam, T.; Lindblom, A.; Schreiber, M.J.; Ling, L.; Chow, A.; Ooi, E.E.; Hibberd, M.L. Characterization of early host responses in adults with dengue disease. BMC Infect. Dis. 2011, 11, 209, doi:10.1186/1471-2334-11-209.
  21. John, D.V.; Lin, Y.S.; Perng, G.C. Biomarkers of severe dengue disease - A review. J. Biomed. Sci. 2015.
  22. Bente, D.A.; Rico-Hesse, R. Models of dengue virus infection. Drug Discov. Today Dis. Model. 2006, 3, 97–103, doi:10.1016/j.ddmod.2006.03.014.
  23. Seneviratne, S.L.; Malavige, G.N.; de Silva, H.J. Pathogenesis of liver involvement during dengue viral infections. Trans. R. Soc. Trop. Med. Hyg. 2006, 100, 608–614, doi:10.1016/j.trstmh.2005.10.007.
  24. Samanta, J. Dengue and its effects on liver. World J. Clin. Cases 2015, doi:10.12998/wjcc.v3.i2.125.
  25. Wu, S.J.L.; Grouard-Vogel, G.; Sun, W.; Mascola, J.R.; Brachtel, E.; Putvatana, R.; Louder, M.K.; Filgueira, L.; Marovich, M.A.; Wong, H.K.; et al. Human skin Langerhans cells are targets of dengue virus infection. Nat. Med. 2000, doi:10.1038/77553.
  26. Noisakran, S.; Onlamoon, N.; Songprakhon, P.; Hsiao, H.M.; Chokephaibulkit, K.; Perng, G.C. Cells in dengue virus infection in vivo. Adv. Virol. 2010, 2010, 164878, doi:10.1155/2010/164878.
  27. Taylor, K.L.; Grant, N.J.; Temperley, N.D.; Patton, E.E. Small molecule screening in zebrafish: An in vivo approach to identifying new chemical tools and drug leads. Cell Commun. Signal. 2010, 8, 11, doi:10.1186/1478-811X-8-11.
  28. Bency, B.J.; Helen, P.A.M. In silico identification of dengue inhibitors in Giloy (Tinospora cordifolia) and Papaya. J. Emerg. Technol. Innov. Res. 2018, 5, 506–511.
  29. Eshun, K.; He, Q. Aloe Vera: A Valuable Ingredient for the Food, Pharmaceutical and Cosmetic Industries - A Review. Crit. Rev. Food Sci. Nutr. 2004, 44, 91–96, doi:10.1080/10408690490424694.
  30. Sharma, N.; Mishra, K.P.; Chanda, S.; Bhardwaj, V.; Tanwar, H.; Ganju, L.; Kumar, B.; Singh, S.B. Evaluation of anti-dengue activity of Carica papaya aqueous leaf extract and its role in platelet augmentation. Arch. Virol. 2019, 164, 1095–1110, doi:10.1007/s00705-019-04179-z.
  31. Houston, D.M.J.; Bugert, J.J.; Denyer, S.P.; Heard, C.M. Potentiated virucidal activity of pomegranate rind extract (PRE) and punicalagin against Herpes simplex virus (HSV) when coadministered with zinc (II) ions, and antiviral activity of PRE against HSV and aciclovir-resistant HSV. PLoS One 2017, 12, e0179291, doi:10.1371/journal.pone.0179291.
  32. Neurath, A.R.; Strick, N.; Li, Y.Y.; Debnath, A.K. Punica granatum (pomegranate) juice provides an HIV-1 entry inhibitor and candidate topical microbicide. Ann. N. Y. Acad. Sci. 2005, 1056, 311–327, doi:10.1196/annals.1352.015.
  33. Tang, L.I.C.; Ling, A.P.K.; Koh, R.Y.; Chye, S.M.; Voon, K.G.L. Screening of anti-dengue activity in methanolic extracts of medicinal plants. BMC Complement. Altern. Med. 2012, doi:10.1186/1472-6882-12-3.

Reviewer 2 Report

  1. Line 53:The author should explain in more detail why there is lack of DENV model from mouse or non-human primates. Commonly speaking, mammals were more relevant to human than zebrafish.
  2. Line 112: 1 ul of serum samples were harvested from carrier fishes, but in line 229,  the volume is 3 ul. how much serum samples were harvested from carrier zebrafish to the study zebrafish?
  3. section 2.6, how did the authors feed the Denguenil to the fish? they said Denguenil were mixed with feed, that situation seemed to let zebrafish eat freely, how did they ensure the three dosages of Denguienil were accurately administrated to the zebrafish in the treatment groups? Furthermore, which one of the three dosages 5.6, 28, 140 ug/kg equals to the human prescribed dose 1000 mg/day, BID? please explain.
  4. Line 260, Fig 1C should be 1B. Line 200, 0.5 gm should be 0.5 mg.
  5. section 3.4, how authors recognized the liver inflammatory cells? any photoes?
  6. In order to reveal the non-toxicity of Denguenil, the authors used cell assays. Why not use Denguenil in the highest dosage (140 ug/kg) to treat healthy zebrafish? This will make a direct comparison of activity and toxicity of Denguienil in the same animal model.  

Author Response

REVIEWER 2:

Comments and Suggestions for Authors

  1. Line 53: The author should explain in more detail why there is lack of DENV model from mouse or non-human primates. Commonly speaking, mammals were more relevant to human than zebrafish.

Response: We thank the reviewer for the valuable suggestion. Development of an appropriate animal model for DENV infection has been hindered by

  • Non-human primates can sustain DENV replication even after inoculation with a dose of 105PFU, the presumed mosquito inoculation in humans [1].
  • lack of clinical disease in non-human primates.
  • While some rhesus macaques occasionally display low platelet counts, they do not display any other overt clinical signs after sub cutaneous DENV infection [2]
  • Three days post infection, monkeys display only few signs of haemorrhage, including petechiae and hematomas, coagulopathy with increased D-dimers related to DIC. However, they do not display any other signs including fever, anorexia or lethargy [3]

Further, lack of a suitable small-animal model of dengue infection has greatly hindered the study of dengue pathogenesis and the development of novel therapeutics [4][5].

  • lack of replication of DENV clinical isolates in wild-type (WT) mice,
  • While DENV tropism has been identified both animal models and humans, the active site of viral replication is still unclear.
  • Mouse models of dengue did not show human clinical signs of Dengue viral infection but developed neurotropic disease, which is not generally observed in humans [6].
  • While few strains of mice support minimal replication, other strains show paralysis [7].
  • In mouse models, DENV infection induces limited DENV- pathogenesis such as liver damage, increased white blood cell (WBC) counts, thrombocytopenia, and an increase in hematocrit level reminiscent of vascular leak [8].

Further, use of zebrafish for disease modelling of rare genetic blood diseases and in drug discovery application in Parkinson’s disease and other movement disorders is apparent [9][10]. Use of zebrafish model of dengue viral pathology would help in early screening of potential anti-dengue drugs and also aid in proof of concept studies thereby, reducing the gap in drug development and increasing overall drug success rates.

This information was captured in the revised introduction part of the manuscript.

  1. Line 112: 1 ul of serum samples were harvested from carrier fishes, but in line 229, the volume is 3 ul. how much serum samples were harvested from carrier zebrafish to the study zebrafish?

Response: We thank the reviewer for the query. We apologize for the typographical error.  Human serum was injected at a volume of 3 µl into zebrafish which were primary carriers of the virus. The virus was allowed to replicated for up to 14 days in the primary carrier zebrafish and 1µl of the serum from a total of 5 primary carriers of zebrafish were harvested and diluted to 100 µl with PBS. From this aliquot a 3 µl was injected into the zebrafish that were part of the study group. This information was included in the revised manuscript section 2.3 and 3.1.

  1. section 2.6, how did the authors feed the Denguenil to the fish? they said Denguenil were mixed with feed, that situation seemed to let zebrafish eat freely, how did they ensure the three dosages of Denguienil were accurately administrated to the zebrafish in the treatment groups? Furthermore, which one of the three dosages 5.6, 28, 140 ug/kg equals to the human prescribed dose 1000 mg/day, BID? please explain.

Response: We thank the reviewer for the valuable query. In the present investigation, fish were feed with the indicated doses individually in separated tanks and once the dosing is complete, they were put back into the original tank for housing.

For fish that are fed with pellets, drug dosage accurately controlled by formulating the size of the pellet and the concentration of the drug in the pellet. Zebrafish was able to easily swallow 4mg of fish feed pellet which is circular in shape and at a diameter of 1mm. The drug was accordingly mixed with the feed. Also, zebrafish fed at the rate of three pellets per day is right dosage for health and immediate consumption. Therefore, the fish swallowed the pellet immediately within 3-5 seconds of dropping the feed in the feed tank. This formulation method prevented the feed from being soaked in the water and the drug eluting out the drug [11] [12]. Further feed grade coconut oil is sprayed on top of the feed after it is dried to offer a coating against denaturation of the drug on exposure to either water or air. In the present investigation, out of the three doses, the mid dose i.e 28 µg/kg is equal to human prescribed dose.  This information was captured in section 3.1 of results.

  1. Line 260, Fig 1C should be 1B. Line 200, 0.5 gm should be 0.5 mg.

Response: We thank the reviewer for the valuable query. We apologise for the typographical error. In the revised manuscript we corrected the Fig 1C to Fig. 1B. We have re-verified for the amount of denguenil used, and we confirm that we have used 0.5 gm denguenil powder for the analysis.

  1. section 3.4, how authors recognized the liver inflammatory cells? any photoes?

Response: We thank the reviewer for the valuable suggestion. In the present investigation, liver inflammatory cells were observed by the presence of cells with large densely stained nucleus, cells are organized as mobile cells and not with an organized pattern. In terms of colour, inflammatory cells are differentiated by their deeper haematoxylin stain and shape established by cell membrane which is different from hepatocytes [13].

  1. In order to reveal the non-toxicity of Denguenil, the authors used cell assays. Why not use Denguenil in the highest dosage (140 ug/kg) to treat healthy zebrafish? This will make a direct comparison of activity and toxicity of Denguenil in the same animal model.  

Response: We thank the reviewer for the valuable suggestion. In the present investigation we have tested the cyto safety of denguenil using only in vitro cell based studies. Zebrafish translational dose from human dose has been established. In the present study, during the course of the study, dosing with denguenil did not show any signs of body weight loss, or others forms of systemic or acute toxicity. Moreover, during screening denguenil treated group did not show any toxicity or organ pathology beyond disease control groups hence we did not consider a toxicity screen.

REFERENCES

  1. Marchette, N.J.; Halstead, S.B.; Falkler, W.A.; Stenhouse, A.; Nash, D. Studies on the pathogenesis of dengue infection in monkeys. III. sequential distribution of virus in primary and heterologous infections. J. Infect. Dis. 1973, doi:10.1093/infdis/128.1.23.
  2. Halstead, S.B.; Shotwell, H.; Casals, J. Studies on the pathogenesis of dengue infection in monkeys. I. clinical laboratory responses to primary infection. J. Infect. Dis. 1973, doi:10.1093/infdis/128.1.7.
  3. Onlamoon, N.; Noisakran, S.; Hsiao, H.M.; Duncan, A.; Villinger, F.; Ansari, A.A.; Perng, G.C. Dengue virus - Induced hemorrhage in a nonhuman primate model. Blood 2010, doi:10.1182/blood-2009-09-242990.
  4. Zompi, S.; Harris, E. Animal models of dengue virus infection. Viruses 2012, 4, 62–82.
  5. Krishnakumar, V.; Durairajan, S.S.K.; Alagarasu, K.; Li, M.; Dash, A.P. Recent updates on mouse models for human immunodeficiency, influenza, and dengue viral infections. Viruses 2019, 11, 252, doi:10.3390/v11030252.
  6. Raut, C.G.; Deolankar, R.P.; Kolhapure, R.M.; Goverdhan, M.K. Susceptibility of laboratory-bred rodent to the experimental infection with dengue virus type 2. Acta Virol. 1996.
  7. Shresta, S.; Kyle, J.L.; Beatty, P.R.; Harris, E. Early activation of natural killer and B cells in response to primary dengue virus infection in A/J mice. Virology 2004, doi:10.1016/j.virol.2003.09.048.
  8. Paes, M. V.; Pinhão, A.T.; Barreto, D.F.; Costa, S.M.; Oliveira, M.P.; Nogueira, A.C.; Takiya, C.M.; Farias-Filho, J.C.; Schatzmayr, H.G.; Alves, A.M.B.; et al. Liver injury and viremia in mice infected with dengue-2 virus. Virology 2005, doi:10.1016/j.virol.2005.04.042.
  9. Vaz, R.L.; Outeiro, T.F.; Ferreira, J.J. Zebrafish as an animal model for drug discovery in Parkinson’s disease and other movement disorders: A systematic review. Front. Neurol. 2018.
  10. Rissone, A.; Burgess, S.M. Rare genetic blood disease modeling in zebrafish. Front. Genet. 2018.
  11. Penglase, S.; Moren, M.; Hamre, K. Lab animals: Standardize the diet for zebrafish model. Nature 2012.
  12. Ulloa, P.E.; Medrano, J.F.; Feijo, C.G. Zebrafish as animal model for aquaculture nutrition research. Front. Genet. 2014, doi:10.3389/fgene.2014.00313.
  13. Conrad, R.; Castelino-Prabhu, S.; Cobb, C.; Raza, A. Cytopathologic diagnosis of liver mass lesions. J. Gastrointest. Oncol. 2013.

Round 2

Reviewer 1 Report

Reviewer point 4

The author enlarged the magnification of Figure 2. However, the position of arrows are shifted if the author compared with previous version.

Author Response

Reviewer point 4

The author enlarged the magnification of Figure 2. However, the position of arrows are shifted if the author compared with previous version.

Response: We thank the reviewer for the valuable suggestion. As suggested, we modified the Fig.2 and made sure that the arrows were in place in the revised manuscript.